

# Rainfall intensity bursts and the erosion of soils: an analysis highlighting the need for high temporal resolution rainfall data for research under current and future climates.

David L. Dunkerley[1]

[1]School of Earth, Atmosphere and Environment, Monash University, Melbourne Victoria, 3800, Australia.

*Correspondence to*: David L. Dunkerley (david.dunkerley@monash.edu)

**Abstract.** Many landsurface processes, including splash dislodgment and downslope transport of soil materials, are influenced
strongly by short-lived peaks in rainfall intensity but are less well accounted for by longer-term average rates. Specifically, rainfall intensities reached over periods of 10-30 minutes appear to have more explanatory power than hourly or longer-period data. However, most analyses of rainfall, and particularly scenarios of possible future rainfall extremes under climate change, rely on hourly data. Using two Australian pluviograph records with 1 second resolution, one from an arid and one from a wet tropical climate, the nature of short-lived intensity bursts is analysed from the raw inter-tip times of the tipping bucket gauges.
Hourly apparent rainfall intensities average just 1.43 mm h$^{-1}$ at the wet tropical site, and 2.12 mm h$^{-1}$ at the arid site. At the wet tropical site, 'intensity bursts' of extreme intensity occur frequently, those exceeding 30 mm h$^{-1}$ occurring on average at intervals of < 1 d, and those of > 60 mm h$^{-1}$ occurring on average at intervals of < 2 d. These bursts include falls of 13.2 mm in 4.4 minutes, the equivalent of 180 mm h$^{-1}$, and 29 mm in 12.6 minutes, equivalent to 138 mm h$^{-1}$. Intensity bursts at the arid site are much less frequent, those of 50-60 mm h$^{-1}$ occurring at intervals of ~ 1 month; moreover, the bursts have a much
shorter duration. The aggregation of rainfall data to hourly level conceals the occurrence of many of these short intensity bursts, which are potentially highly erosive. A short review examines some of the mechanisms through which intensity bursts affect infiltration, overland flow, and soil dislodgment. It is proposed that more attention to resolving these short-lived but important aspects of rainfall climatology is warranted, especially in light of possible changes in rainfall extremes under climate change.

## 1. Introduction

Soil erosion is an important risk factor for future pastoral and agricultural production, as well as water quality, and is considered highly likely to be affected by climate change (Nearing et al. 2004, Klik & Eitzinger 2010, Mullan et al. 2012,
Segura et al. 2014, Garbrecht et al. 2015, Mondal et al. 2015, Sharratt et al. 2015, Li & Fang 2016, Giang et al. 2017). Soil erosion consequently bears on global food security, in addition to the more direct effects of changing climate on crop yields (Rhodes 2014), and a need for adaptations to protect agricultural productivity has been highlighted (Blanco & Lal 2008). The



erosion of soils by rainsplash and flowing water involves a multitude of processes acting across spatial and temporal scales, from momentary splash dislodgment of a small particle to sediment remobilisation from temporary storage in locations such as alluvial fans, footslopes or floodplains. There are many reasons for seeking to understand and track soil erosion in the contemporary landscape, and to work towards an understanding of how it may respond to climate change. The burgeoning

human population and increased need for food, fuel and fibre is placing increasing demands on cultivated and rangeland soils, for which some contemporary erosion rates are already considered unsustainable (Panagos et al. 2015). Simultaneous changes in global and regional hydroclimates, related to global warming and shifts in the nature of the hydrologic cycle, may result in increased frequency and erosivity of rainfall events (Hatfield et al. 2013), as well as related ecosystem changes including in floristics, plant architecture, and soil moisture content. At the same time, growth in urban areas and populations, and the

associated impervious urban land surfaces, appears to be resulting in increased risk of urban flash flooding and associated sediment transport, especially in peri-urban areas (Esposito et al. 2018), though this may be partially offset by the growing adoption of 'sponge city' and similar approaches to building more absorbent cities (Li et al. 2017, Li et al. 2018, Dong et al. 2018). Archer & Fowler (2018) have illustrated the effects of short-term intensity bursts in UK flooding, emphasising that short-period intensity was more significant than total rainfall. Additionally, there are indications that fire regimes, especially

the occurrence of large wildfires, will be altered by climate change (van Bellen et al. 2010, Bowman et al. 2013, Harvey 2016). As will be discussed below, post-fire landscapes are especially vulnerable to intense soil erosion. In addition to these considerations and others that could be listed, soils themselves may alter in response to climate change, with changes in organic matter content and the activity of soil biota possibly modifying soil structure, aggregation, and erodibility (Karmakar et al. 2016). It is clear, therefore, that multiple factors may result in different rates or mechanisms of soil erosion in coming decades,

posing a significant challenge for the informed management of agricultural and other soils.

Extreme rainfalls, especially at short timescales of minutes or tens of minutes, are widely recognised as important drivers of soil erosion. Some of the field evidence will be summarised later. Despite this, they have received less attention than more lumped measures of rainfall, such as daily totals, from which change detection is undertaken by such methods as tallying the number of daily rainfalls exceeding the historic 95[th] percentile (or some other statistic) of daily rainfalls (Schär et al. 2016).

For example, in comparing 20[th] Century data with model scenarios for 2081-2100, O'Gorman & Schneider (2009) employed the 99.9[th] percentile daily rainfall as a measure of extreme rainfall. They report that in the tropics, 20[th] Century extreme daily rainfalls exceeded 50 mm, but in model scenarios for the late 21[st] C this increased to about 70 mm (O'Gorman & Schneider 2009 Figure 1). Whilst extreme daily totals such as these are an informative measure for the purposes of climate science, in the context of soil erosion, they do not provide sufficient resolution to understand erosional events. Soil loss is often most

successfully accounted for by measures of sub-daily rainfall intensity over minutes or tens of minutes (see below). Attempts to explore sub-daily extreme rainfalls are hampered because many available rainfall data sets, as well as scenarios of future rainfalls under climate change, commonly provide only daily or hourly resolution (Guerreiro et al. 2018). In analysing sub-daily rainfall data from 5347 stations globally, Monjo (2016) reported that only 17.8% had a reporting frequency of 1 h, and





about 75% reported rainfalls 2-4 times per day, 42.2% having a 6 h resolution. Some workers have focussed on common, rather than extreme, rainfalls. Pendergrass & Deser (2017) for instance argued that ordinary rainfalls deliver most rain and release most latent heat and are thus more relevant to climate studies. Studies of rainfall extremes employing daily rainfall amounts include Donat et al. (2013) and Keggenhoff et al. (2014); studies of extremes relying on hourly totals include Sun et

al. (2017) who analysed rainfall extremes over Eastern China, Cortés-Hernández et al. (2016) for eastern Australia and Beranova et al. (2018) for the Czech Republic. Similarly, many studies of the secular trends in extreme events rely on hourly or daily rainfall depth data (Costa & Soares 2008; Peralta-Hernandez 2009; Formayer & Fritz 2017; de Waal et al. 2017; Lupikasza et al. 2017, Yu et al. 2017). Yet other studies rely on data aggregated to longer periods, such as the 2-3 h aggregation used by Luo & Wang (2015). There are exceptions to the general reliance on daily and hourly rainfall data, which include

Yilmaz & Perera (2015), who examined 10 min and 30 min data, as well as hourly and longer aggregations, in a study of extreme rainfalls in Victoria, Australia. Many studies of rainfall occurrence, including extremes, consequently use parameters selected under constraints of data availability. These may involve average intensities tallied using data pooled from many rain days, and which conceal variation among days. For instance, Polemio & Lonigro (2015) used a 'monthly rainfall intensity' which is the monthly rainfall depth scaled by the number of rain days in that month. Others use a similar 'daily rain intensity'

(e.g. Nandargi & Mulye 2012) or the 'simple daily intensity index', which is defined as the ratio of the annual total precipitation to the number of wet days (Hatzaki et al. 2010, Zhang et al. 2011, Gao et al. 2018). Many such indices, with their very limited temporal resolution, are likely to offer restricted explanatory power in relation to splash and water erosion of soils or to other landsurface hydrological processes.

20       Motivated by the foregoing, the goal of the present paper is to present a brief overview of short-term rainfall extremes, focussing primarily on what are here referred to as 'intensity bursts': short periods of intense rainfall contained within longer events of lower overall intensity. This terminology echoes that of Peters and Christensen (2002), who explored the burst-like behaviour of rainfall, which they likened to the sporadic temporal occurrence of earthquakes or mass movements. To support the objective of highlighting the nature and importance of these short-duration bursts, a brief review is provided of some of

the mechanisms through which intensity bursts drive processes such as surface ponding and the splash dislodgment of soil particles. However, as a first step, using rainfall data from the Australian desert and wet tropics, intensity bursts are explored quantitatively using tipping bucket raingauge data having 1 sec temporal resolution. Intensity is assessed from the time between successive tipping bucket tip events, with no aggregation. It is the aim of this short overview to highlight the need for more study of rainfall behaviour at high temporal resolution, especially to characterise intensity bursts in a way that will facilitate

further exploration of their influence on surface processes. There is currently little or no literature exploring the likely changes in short-lived intensity bursts under future climate change. In a sense this is unsurprising, as there has been little investigation of intensity bursts even in empirical data collected under the present climate. Nevertheless, it will be argued here that short-term intensity bursts warrant further consideration, especially in relation to surface hydrology and soil erosion.



## 1.1 What is meant by 'intensity burst'?

It is appropriate to begin by further describing intensity bursts. The analysis presented below seeks both to illustrate the nature of these bursts at two field locations, and to highlight the important differences between the kinds of indices
commonly adopted in climate science and those employed in research on soil erosion and related landsurface processes. The expression 'intensity burst' is used here without a rigorous, formal definition, to refer to periods of very intense rain that occupy a small fraction of the duration of a longer enclosing event in which the typical intensity is lower (Figure 1). The literature provides no established definitions with which to define an intensity burst. Here, the use of a threshold intensity of 30 mm h$^{-1}$ is adopted. This is the intensity above which rainfall intensity is classified as "extreme" by Tokay & Short (1996).
The contrast between the mean intensity of rainfall events and the intensity of bursts that occur within them can be marked. The intensity of the bursts shown in Figure 1 and the mean intensity of the rainfall events within which they occurred are set out in Table 1.

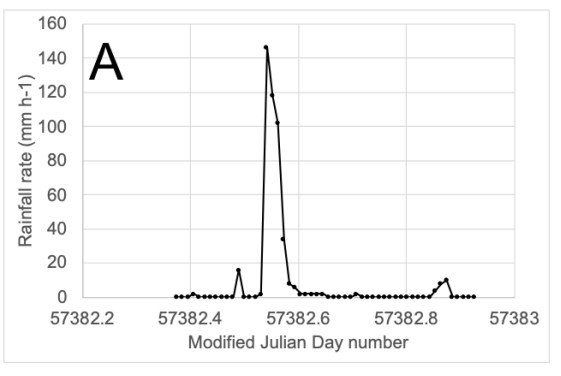
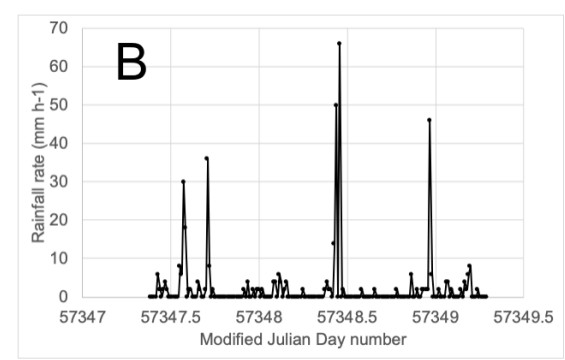
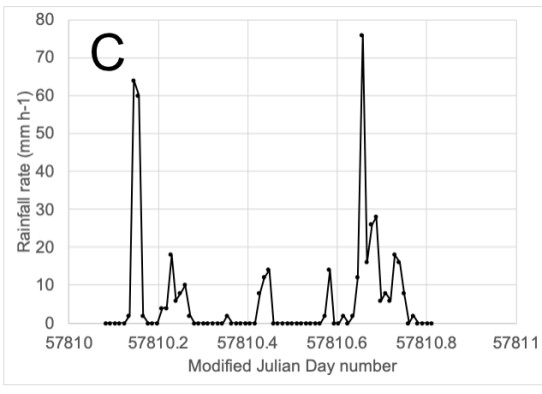
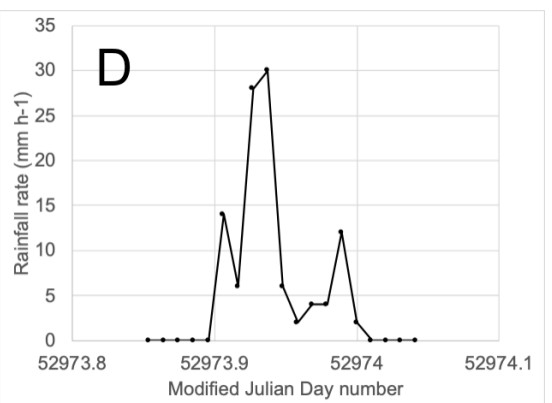

**Figure 1** Examples of intensity bursts in rainfall records from the field sites, shown using data aggregated to 15-minute totals. Each dot represents a 15 min period. Field locations: A- C: MM D: FG. The four intensity plots shown here each show the



entirety of a rainfall event defined using MIT = 6 h (refer to text for details). Each event begins with the first rainfall and ends with the last rainfall; preceding and subsequent observations showing zero rainfall rate are not included in the events. Event A is mostly rainless but includes a single very intense burst; events B and C contain one or more bursts together with periods of lower intensity; event D shows continuous rainfall with a central intensity burst.

| location | Start date of rainfall event (MJD) | End date of rainfall event (MJD) | Duration of event (h) | Depth of event (mm) | Mean rainfall intensity of event (mm h$^{-1}$) | Peak intensities of burst(s) within event from 15 min data (mm h$^{-1}$) |
|---|---|---|---|---|---|---|
| MM | 57382.40 | 57349.24 | 13.9 | 42.8 | 1.32 | 146 |
| MM | 57347.42 | 57349.24 | 43.6 | 46.8 | 0.98 | 30, 36, 67, 48 |
| MM | 57810.12 | 57810.77 | 15.4 | 45.8 | 2.97 | 64, 76 |
| FG | 52973.90 | 52973.99 | 2.3 | 27.0 | 11.63 | 30 |

**Table 1** Details of rainfall events defined using MIT = 6 h, together with the mean rainfall intensity of the event and the peak burst intensities evident in data aggregated to 15 min rainfall totals. See text for details.

## 2. Field sites and methods of data collection and processing

2.1 Field locations and the tipping-bucket pluviograph data

Data from two contrasting field locations are analysed here; one arid and one wet tropical. The wet tropical field location was near the township of Millaa Millaa, on the Atherton Tablelands, within the wet tropics of far northern Queensland, Australia. The site (hereafter designated MM) was located at 650 m above sea level, and 40 km inland from the Coral Sea coastline. Local mean annual rainfall is estimated to be ~ 3 m, primarily falling in a wet season from January to May. The rain
is orographically-enhanced in moist SE trade winds that rise over rugged uplands immediately inland of the coast. The data analysed here were recorded during 2014-2017 and yielded a total of 9.15 m of rainfall (and > 45,000 tipping bucket events). The dryland site was located on the Fowlers Gap Arid Zone Research Station (hereafter FG), located about 100 km north of the regional city of Broken Hill, in the arid far west of New South Wales, Australia. Annual rainfall averages 220 mm but



varies markedly from year to year, related to the ENSO cycle of drought and wet years. Some aspects of the rainfall climatology of this site, including the nature of rainfall in wet and dry years and the intensity variations at intra-event timescales, have been described previously (Dunkerley 2010, 2013). Fowlers Gap is located ~ 950 km from the Australian east coast, and more than 450 km from the nearest coastline in Spencer Gulf, South Australia. The continuous record from this site was collected in the

period 2002-2012, and records about 2.6 m of rainfall (~ 5300 bucket tip events).

The raw data analysed here consists of 0.2 mm tipping bucket events (MM) and 0.5 mm events (FG), logged with a 1 second resolution and analysed without any time aggregation. There were no missing data from either rainfall record. The data logger files of bucket tip events that were recorded in the Gregorian calendar were converted to Modified Julian Days

(MJD), using double-precision FORTRAN routines from the International Astronomical Union's 'SOFA' (Standards of Fundamental Astronomy) subroutine library (http://www.iausofa.org). The use of Modified Julian Days, which begin at midnight rather than noon as for Julian days, facilitated the tallying of daily totals, and Julian dates in general allow the length of rain events or bursts to be found simply by subtracting the starting and ending Julian date. The Modified Julian Day subroutine from SOFA (CAL2JD) was modified to include time with a resolution of ~1 s. Hourly and daily rainfalls were also

extracted, as well as counts of rain days and rainless days. Rainfall events were also identified from the MJD files, using the minimum inter-event time (MIT) approach (Dunkerley 2008) with a 6 h MIT.

2.2 Brief descriptions of the rainfall characteristics at the MM and FG field sites

At MM, the discrete rainfall events defined by the 6 h MIT (N=652) lasted on average 18.6 h, and delivered 21.3 mm at an average intensity of 2.2 mm $h^{-1}$. The average rain day (day with at least one bucket tip, i.e., ≥0.2 mm) amount was 11.7 mm, yielding an average daily mean intensity of 0.48 mm $h^{-1}$. From a corresponding analysis of rain hours (again, hours with ≥ 0.2 mm, N= 6409 hours), the average hourly rainfall was 1.43 mm (average hourly intensity 1.43 mm $h^{-1}$). The 95th, 99th, and 99.9th percentiles of hourly rainfall intensity were 5 mm $h^{-1}$, 13 mm $h^{-1}$, and 32.1 mm $h^{-1}$. The maximum hourly rainfall

was 60.6 mm. At a daily scale, the rain was intermittent, and on average, 66% of the hours on a rain day were rainless, and this accounts for the difference between daily and hourly mean rainfalls. These figures suggest that the rainfall climatology of the MM study site is characterised by quite low intensities, falling in events that typically last less than a day. However, aggregating the data to daily or hourly level involves sacrificing the resolution of the raw bucket tip data; the data on intensity bursts, examined shortly, present quite a different description of the rainfall.

At FG, the average rain day (day with at least one bucket tip, i.e., ≥0.5 mm) amount was 7.58 mm, yielding an average daily mean intensity of 0.32 mm $h^{-1}$. For rain hours (N=1259 hours), the mean amount was 2.12 mm (average hourly intensity 2.12 mm $h^{-1}$). The 95th, 99th, and 99.9th percentiles of hourly rainfall intensity were 8 mm $h^{-1}$, 14.5 mm $h^{-1}$, and 26.1 mm $h^{-1}$. The maximum hourly rainfall was 23.5 mm. In total, 85.1% of hours in the record were rainless. The fact that the hourly



rainfall at FG was 6.6 times larger than the daily intensity reflects the high intermittency of rain during a rain day. At MM, the hourly rainfall was only 2.9 times larger than the daily intensity, reflecting lower intermittency of rain on rain days at the wet tropical site.

In summary, average wet day rainfalls and average daily intensities were both larger at MM. However, average wet hour rainfalls were larger at FG. This reflects the shorter duration and lesser intermittency of rain in the arid conditions at FG. Thus, the level of aggregation of the rainfall data – daily or hourly – affects the apparent intensity of the rainfall. This effect is of considerable potential importance when seeking to understand soil erosion and landsurface hydrology. Moreover, actual intensities during rainfall, assessed over sub-daily durations that involve less loss of resolution, reveal quite different intensities
than those just reviewed, as will be shown next.

     In what follows, the intensity was calculated with minimal loss of resolution from the raw, unaggregated inter-tip times (ITTs) between bucket tips (the time taken for 0.2 mm or 0.5 mm of rain to be recorded). During long ITTs, reflecting low-intensity rainfall, there may be fluctuations in intensity or periods of complete cessations of rainfall as the bucket
progressively fills. In contrast, during the short ITTs associated with extreme intensity bursts, it is possible to be confident that the intensity calculated from the ITT is indicative of the rainfall intensity as the bucket filled through seconds or minutes. Intensities were extracted from the data logger files for ITTs in the range 5 min – 0.1 min. The ITT indicative of at least 30 mm h$^{-1}$ at FG is 1 min, and at MM, 0.4 min. The sequences of time-varying ITTs provide an indication of the intensity of rain from moment to moment. However, the character of intensity bursts was assessed by examining the lengths of uninterrupted
sequences of short ITTs whose duration did not exceed a nominated value, such as 30 sec. Such runs of short ITTs represent sustained intensities exceeding a threshold intensity determined by the nominated ITT. For purposes of reporting, these sequences are here termed 'runs' of short ITTs. The durations of all runs for ITTs of up to 5 min were extracted from the data for MM and FG. Tables 2 and 3 show the minimum intensity required to result in runs of short ITTs of specified duration for the two field sites. Thus, at MM, runs of tips occurring at least every 60 s represent rainfall of at least 12 mm h$^{-1}$ ('very heavy'
rainfall, according to the classification of Tokay & Short 1996). Runs of tips occurring at least every 30 s would represent rainfall of at least 24 mm h$^{-1}$, and tips occurring at least every 15 sec correspond to intensities of at least 48 mm h$^{-1}$.

| Millaa Millaa (MM). Tipping bucket capacity: 0.2 mm | | | | | | |
|---|---|---|---|---|---|---|
| ITT (minute) | Minimum average intensity (mm h$^{-1}$) | Number of bucket tips per h at minimum intensity | Number of tips in longest run of events | Duration of longest run (minute) | Rain depth in longest run (mm) | Observed average intensity of longest run (mm h$^{-1}$) |



| 5 | 2.4 | 12 | 673 | 322.6 | 134.6 | 25.0 |
|---|---|---|---|---|---|---|
| 4 | 3.0 | 15 | 671 | 316.7 | 134.2 | 25.4 |
| 3 | 4.0 | 20 | 664 | 300.4 | 132.8 | 26.5 |
| 2 | 6.0 | 30 | 452 | 183.6 | 90.4 | 29.5 |
| 1 | 12.0 | 60 | 313 | 88.5 | 62.6 | 42.4 |
| 0.8 | 15.0 | 75 | 311 | 63.1 | 62.2 | 59.1 |
| 0.6 | 20.0 | 100 | 309 | 61.8 | 61.8 | 60.0 |
| 0.5 | 24.0 | 120 | 270 | 49.4 | 54.0 | 65.6 |
| 0.4 | 30.0 | 150 | 149 | 22.5 | 29.8 | 79.5 |
| 0.3 | 40.0 | 200 | 146 | 17.2 | 29.2 | 101.9 |
| 0.25 | 48.0 | 240 | 145 | 12.6 | 29.0 | 138.1 |
| 0.2 | 60.0 | 300 | 143 | 11.7 | 28.6 | 146.7 |
| 0.1 | 120.0 | 600 | 66 | 4.4 | 13.2 | 180.0 |

**Table 2** Details of the intensity that would be achieved in runs of ITTs of fixed duration from 0.1 min to 5 min (columns 1 and 2), together with the number of bucket tip events that would result in 1 hour for each ITT (column 3). The remaining columns present the observed data for MM. Note that for every ITT, the observed intensity (column 7) is higher than that shown in column 2.

| Fowlers Gap (FG). Tipping bucket capacity: 0.5 mm | | | | | | |
|---|---|---|---|---|---|---|
| ITT (minute) | Minimum average intensity (mm h$^{-1}$) | Number of bucket tips per h at minimum intensity | Number of tips in longest run of events | Duration of longest run (minute) | Rain depth in longest run (mm) | Observed average intensity of longest run (mm h$^{-1}$) |
| 5 | 6.0 | 12 | 81 | 134.9 | 40.5 | 18.0 |
| 4 | 7.5 | 15 | 72 | 130.8 | 36.0 | 16.5 |
| 3 | 10.0 | 20 | 72 | 58.1 | 36.0 | 37.2 |
| 2 | 15.0 | 30 | 71 | 53.6 | 35.5 | 39.8 |
| 1 | 30.0 | 60 | 58 | 21.3 | 29.0 | 81.7 |
| 0.8 | 37.5 | 75 | 50 | 15.9 | 25.0 | 94.5 |
| 0.6 | 50.0 | 100 | 49 | 15.3 | 24.5 | 96.3 |
| 0.5 | 60.0 | 120 | 48 | 14.7 | 24.0 | 98.2 |



| 0.4 | 75.0 | 150 | 24 | 6.7 | 12.0 | 107.8 |
|-----|------|-----|----|-----|------|-------|
| 0.3 | 100.0 | 200 | 17 | 4.1 | 8.5 | 124.7 |
| 0.25 | 120.0 | 240 | 9 | 1.9 | 4.5 | 138.5 |
| 0.2 | 150.0 | 300 | 2 | 0.3 | 1.0 | 176.4 |
| 0.1 | 300.0 | 600 | - | - | - | - |

**Table 3** Details of the intensity that would be achieved in runs of ITTs of fixed duration from 0.1 min to 5 min (columns 1 and 2), together with the number of bucket tip events that would result in 1 hour for each ITT (column 3). The remaining columns present the observed data for FG. Note that for every ITT, the observed intensity (column 7) is higher than that shown in column 2.

The lengths of runs of short ITTs were extracted from the unaggregated bucket tip records. From these data, the relationships among the duration, depth, and intensity of the intensity bursts were analysed. The start and end dates of each burst were recorded from the MJD data, and from these data the frequency of occurrence of bursts at the two field sites, and the length of the time between the occurrence of successive bursts, were recorded.

## 3. Results

At MM, of the 45,737 ITTs, 94.4% were shorter than 60 minutes, 70.0% were shorter than 5 minutes, and 36.5% shorter than 1 minute (Tables 4 and 5). The median ITT was 1.8 minutes, corresponding to an intensity of 6.7 mm h$^{-1}$. At FG, of the 5353 ITTs, 88.0% were shorter than 60 minutes, 49% shorter than 5 minutes, and the median value was 5.2 minutes, corresponding to an intensity of 5.7 mm h$^{-1}$. The median ITTs thus indicate intensities that are 4.7 x larger (MM) and 2.7 x larger (FG) than the average rain hour intensity reported above, which was in turn 2.9-6.6 times larger than the daily intensities. Unsurprisingly, the extent of temporal aggregation results in wide variations in the apparent intensity of the rainfall. This is illustrated in Figure 2, which compares the rainfall rates determined from the unaggregated ITTs with the same data but aggregated to 15 min and 1 h totals, from a single MM rainfall event defined using MIT = 6 h. The event lasted 43.6 h and delivered 46.8 mm of rain at an average intensity of 0.98 mm h$^{-1}$. The raw ITT data show intensity bursts of up to 240 mm h$^{-1}$, whereas the 15 min data show bursts of only about 67 mm h$^{-1}$, or less than one-third as intense; the 1 h data show the largest burst at only 13 mm h$^{-1}$. Moreover, the first two intensity peaks show the effect of burst duration, the first peak being more intense in the raw ITT data and the 1 h data, but the second (shorter) burst is more intense in the 15 min data.





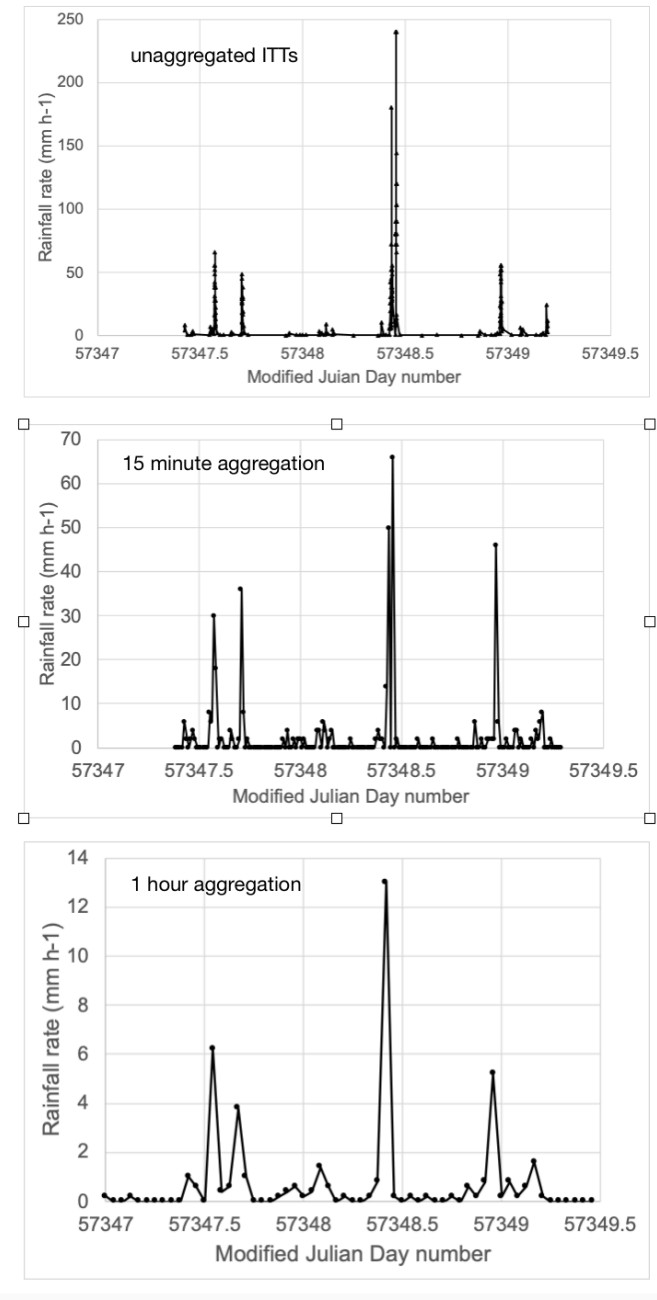

**Figure 2** Effect of time aggregation on the appearance of intensity bursts in the MM data, for a rainfall event starting at Modified Julian Day 57347.42 and ending at 57349.24. Upper: unaggregated ITTs. Middle: data aggregated to 15 min rainfall totals. Lower: data aggregated to 1 h totals. Note that the scale of rainfall intensities differs among the three graphs, owing to the more intense bursts evident in the less aggregated data.



| Length of ITT (mins) | Number of ITTs in record | % of all ITTs |
|---|---|---|
| 0-1 | 16700 | 36.5 |
| 0-5 | 32029 | 70.0 |
| 5-10 | 4663 | 10.1 |
| 10-15 | 1971 | 4.3 |
| 15-20 | 1261 | 2.8 |
| 20-25 | 858 | 1.9 |
| 25-30 | 590 | 1.3 |
| 30-35 | 470 | 1.0 |
| 35-40 | 404 | 0.9 |
| 40-45 | 315 | 0.7 |
| 45-50 | 261 | 0.6 |
| 50-55 | 226 | 0.5 |
| 55-60 | 165 | 0.4 |

**Table 4** Counts of ITTs in the rainfall record from MM, for durations up to 60 minutes, together with the proportion of all tip events falling into each ITT class. The ITTs listed account for 94.4% of all ITTs, the remaining values primarily reflecting long ITTs during gaps between rainfall events.



| Length of ITT (mins) | Number of ITTs in record | % of all ITTs |
|---|---|---|
| 0-1 | 993 | 18.6 |
| 0-5 | 2624 | 49.0 |
| 5-10 | 941 | 17.6 |
| 10-15 | 431 | 8.1 |
| 15-20 | 204 | 3.8 |
| 20-25 | 141 | 2.6 |
| 25-30 | 117 | 2.2 |
| 30-35 | 70 | 1.3 |
| 35-40 | 52 | 1.0 |
| 40-45 | 46 | 0.9 |
| 45-50 | 30 | 0.6 |
| 50-55 | 31 | 0.6 |
| 55-60 | 26 | 0.5 |

**Table 5** Counts of ITTs in the rainfall record from FG, for durations up to 60 minutes, together with the proportion of all tip events falling into each ITT class. The ITTs listed account for 88.1% of all ITTs, the remaining values primarily reflecting long ITTs during gaps between rainfall events.

In terms of the development of surface ponding, overland flow, and the splash dislodgment of soil, runs of short ITTs represent periods of sustained high intensity - an intensity burst. Details of runs of short ITTs corresponding to intensities of at least 30 mm h$^{-1}$ ('extreme' intensity) at both field sites are presented in Table 6.

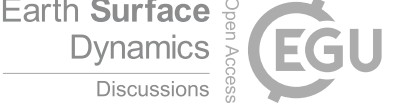

**Millaa Millaa (MM)**

| ITT (min) | N | Average number of ITTs per run | Max number of ITTs per run | Average duration of run (min) | Max duration of run (min) | Average intensity of runs (mm h$^{-1}$) | Average time between commencement of runs (hour) |
|---|---|---|---|---|---|---|---|
| 0.4 | 1854 | 4.5 | 149 | 1.08 | 22.5 | 44.9 | 15.6 |
| 0.3 | 1400 | 4.4 | 146 | 0.88 | 17.2 | 54.4 | 20.6 |
| 0.25 | 1112 | 4.3 | 145 | 0.77 | 12.6 | 62.7 | 26.1 |
| 0.2 | 695 | 3.8 | 143 | 0.54 | 11.7 | 80.5 | 39.1 |
| 0.1 | 286 | 1.45 | 66 | 0.097 | 4.4 | 196.0 | 92.3 |

**Fowlers Gap (FG)**

| ITT (min) | N | Average number of ITTs per run | Max number of ITTs per run | Average duration of run (min) | Max duration of run (min) | Average intensity of runs ( mm h$^{-1}$) | Average time between commencement of runs (day) |
|---|---|---|---|---|---|---|---|
| 1.0 | 217 | 5 | 58 | 2.71 | 21.3 | 45.2 | 15.8 |
| 0.8 | 184 | 5 | 50 | 2.15 | 15.9 | 52.1 | 18.7 |
| 0.6 | 133 | 4 | 49 | 1.57 | 15.3 | 65.8 | 25.4 |
| 0.5 | 92 | 4 | 48 | 1.45 | 14.7 | 75.9 | 36.9 |
| 0.25 | 27 | 2 | 9 | 0.50 | 1.95 | 134.4 | 133.1 |

5  **Table 6** Details of runs of short ITTs forming intensity bursts at the two field sites. The ITT values listed in column 1 were selected to represent intensities of approximately 30, 40, 50, 60, and 120 mm/h. The ITT durations, and the numbers of ITTs shown in column 4, differ between the sites owing to the 0.2 mm sensitivity of the tipping bucket gauge at MM and the 0.5 mm sensitivity at FG. Note also that the average time between the commencement of runs of short ITTs in column 8 are expressed in hours for MM but in days for FG.



Some remarkable runs of short ITTs at MM included 270 successive bucket tips less than 30 s apart. The intensity burst lasted for 49.4 min, delivering 54 mm, the equivalent of a mean intensity of 65.6 mm h$^{-1}$. The longest run of ITTs shorter than 15 s lasted for 12.6 minutes, delivering 29 mm in 145 successive tips, the equivalent of 138 mm h$^{-1}$. The longest run of tips occurring at least every 6 seconds, representing an intensity of at least 120 mm h$^{-1}$, consisted of 66 successive tip events, delivering 13.2 mm in 4.4 minutes, the equivalent of 180 mm h$^{-1}$. Clearly, within these bursts, there must occur sub-periods where the intensity is higher and the ITT shorter than indicated by the threshold ITT adopted to delineate them, since in all cases the calculated intensity of the recorded bursts exceeds the theoretical minimum intensity as set out in Tables 4 and 5. Figure 3 presents the observed intensity data for ITTs up to 5 minutes for both MM and FG, and it is evident that these exceed the nominal minimum intensity by an amount that increases for shorter ITTs. At MM the difference is about 20 mm h$^{-1}$ for ITTs of more than ~2 min, but increases to about 90 mm h$^{-1}$ for ITTs of < 15 seconds, and a similar pattern is evident for FG.

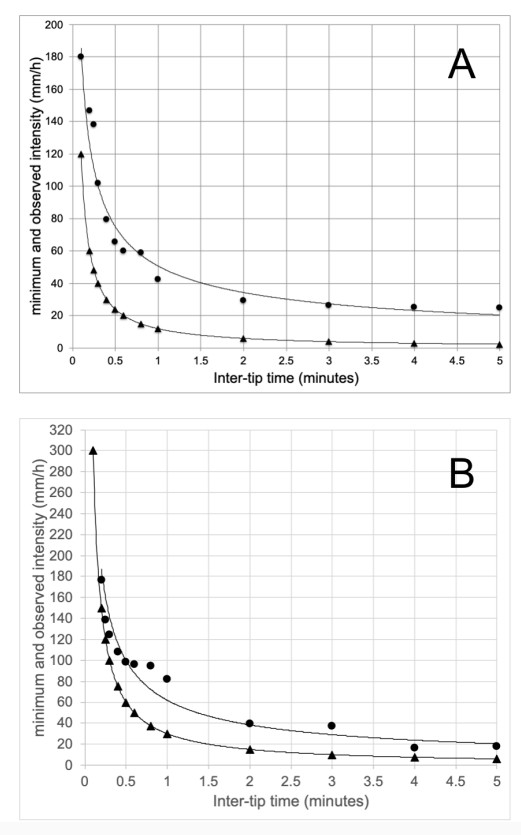

**Figure 3** Expected intensity of intensity bursts of fixed ITTs having varying durations from 0.1 min to 5 min (blue diamond symbols) and observed intensity of longest run of ITTs not exceeding each ITT (red square symbols). Locations: MM (A) and FG (B). Fitted regression relations are shown by solid lines (for equations, see text). Note that in all cases, the observed



intensities exceed those expected from fixed ITTs, indicating that the rainfall included some ITTs that were of shorted duration than the ITT indicated on the X axis.

The mathematical relationship between runs of unvarying ITTs (in minutes) and the resulting rainfall intensity for MM can be described by the exact power function

Intensity (mm h$^{-1}$) = 12 ITT$^{-1.0}$   ……………………….. [1]

For the relationship between the longest runs whose varying ITTs do not exceed nominated ITTs up to 5 min, the
observed relationship for MM becomes

Intensity (mm h$^{-1}$) = 50.73 ITT$^{-0.563}$ ……………………….. [2]

for which ITT is again expressed in minutes, and $r^2 = 0.96$. For FG the corresponding relationship is

Intensity (mm h$^{-1}$) = 61.88 ITT$^{-0.687}$ ……………………….. [3]

with $r^2 = 0.94$.

In terms of the longest bursts comprised of ITTs of less than 5 minutes, Figure 4 shows that the intensity declines for longer runs, which are found for the larger values of ITT. The relationship for MM can be expressed as the power function

Intensity of longest run = 423.6 (duration of longest run)$^{-0.49}$   ……   [4]
in which intensity is in mm h$^{-1}$, the duration of the longest run is expressed in minutes, and $r^2 = 0.98$. For FG, the corresponding relationship is

Intensity of longest run = 199.4 (duration of longest run)$^{-0.40}$   ……   [5]
for which $r^2 = 0.80$.





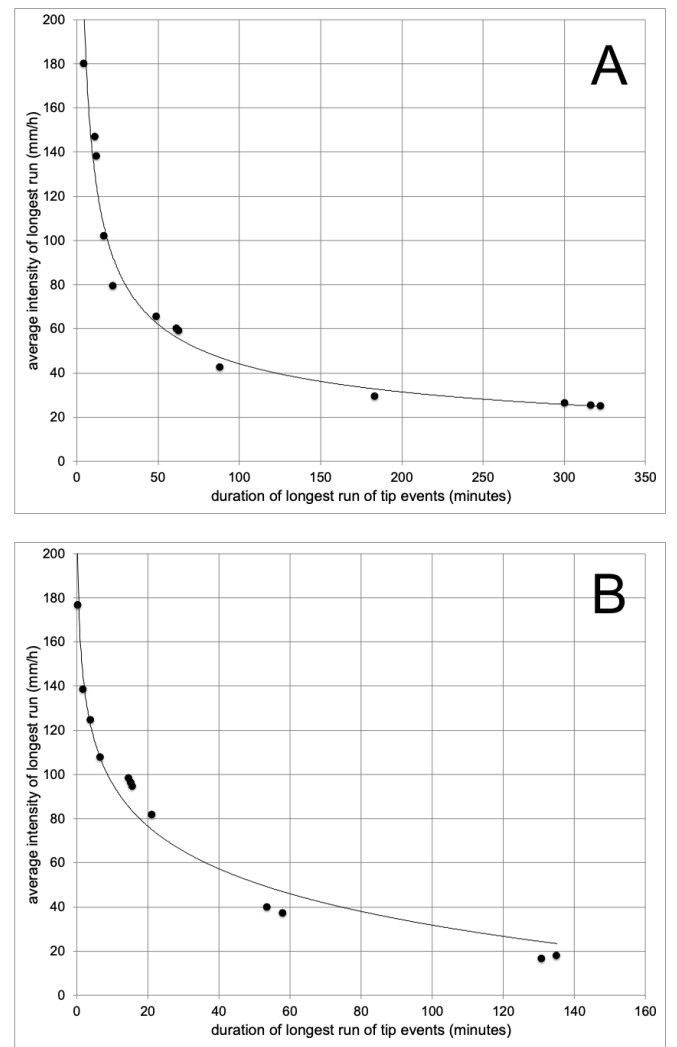

**Figure 4** Relationships between the average intensity of the longest run of ITTs having durations from 0.1 min to 5 min, and the duration of the longest such run. Locations: A: MM B: FG. The solid lines are fitted regression equations (see text for details). Note the rapid rise in intensity for bursts having durations less than about 20 minutes.

3. 1 Is the nature and occurrence of intensity bursts different at the two field sites?

Table 6 shows that there are some notable similarities and differences between intensity bursts at the FG arid site and wet tropical MM. Maximum run durations for bursts exceeding 30 mm h$^{-1}$ are comparable at the two sites – about 20 minutes. Burst durations for higher intensities become shorter, declining to 12-15 minutes for bursts of > 60 mm h$^{-1}$. This is extremely



intense rainfall. Despite the similarities of the bursts, their occurrence is very different. Table 7 shows the average time between the commencement of bursts. For example, bursts exceeding 60 mm h$^{-1}$ at MM occur on average 39.1 h apart (1.6 days). In contrast, at FG events exceeding 60 mm h$^{-1}$ are much less common, the average time between them being 36.9 days. The most intense bursts tabulated in Table 7, those exceeding 120 mm h$^{-1}$, occur on average about every 4 days at MM, but are much

less frequent at FG, where they occur on average every 133 days (fewer than 3 occurrences per year). The other notable difference related to the climate of the two sites is the duration of intensity bursts. These are generally considerably longer at MM than at FG. For instance, bursts of ~ 60 mm h$^{-1}$ last for up to 60 minutes at MM, but only for about 20 minutes at FG.

**4. Discussion highlighting role of intensity bursts in landsurface processes**

    The results just presented demonstrate the occurrence of bursts of rain many times more intense than hourly means – 'intensity bursts' – and reaching intensities of 130-180 mm h$^{-1}$ over durations of approximately 5-15 minutes, and in some cases longer. Runs of ITTs of up to 5 min duration have durations of hours and are striking features of the rainfall climatology,

but are too long to be considered 'bursts' in the sense used here. Nevertheless, many are classified as showing 'extreme' intensities (Tokay & Short 1996). These data can be compared with the average rain day intensity of 0.48 mm h$^{-1}$ or the average rain hour intensity of 1.43 mm h$^{-1}$ noted earlier for MM. The intensity bursts are the kinds of short periods of intense rain that soil erosion researchers have sought to characterise by using fixed clock-period parameters such as $I_{30}$ and $I_{15}$. Given that they have been used to successfully account for soil erosion (see later discussion), intensity bursts evidently deliver sufficiently

intense rain to deepen surface ponding, and so intensify soil splash dislodgment, establish overland flow paths by scouring the soil surface, and drive other surface processes explored below. The burst intensities are in some cases more than two orders of magnitude higher than even the mean rain hour intensity. The characteristics of intensity bursts in environments other than those studied here, such as those experiencing convective storms or frontal rain, appear to be largely unexplored. The intensity bursts have only been partially characterised here for illustrative purposes. A fuller analysis would consider their position

within a rainfall event – early, middle, or late in the event, what relatiobship the bursts bear to the properties of the enclosing event, such as its total duration, depth, or mean intensity, and the frequency with which multiple bursts occur within an enclosing rainfall event. Some preliminary analyses of this kind were presented by Dunkerley (2013).

    There are differences between indices such as $I_{30}$, which are derived from nominated clock periods, and measures

based on runs of ITTs, which can begin and end at any instant, and which can therefore have varying durations. The two measures yield results that are broadly comparable, though there are some significant differences. As an example, the observed $I_{30}$ in the rainfall record at MM was 40.2 mm h$^{-1}$, whereas equation [3] above suggests that an intensity burst of 30 minutes would have an intensity of 79.7 mm h$^{-1}$. To the writer's knowledge, no attempts have been made to relate measures of soil erosion to the duration and intensity of intensity bursts; rather, fixed clock periods such as $I_{30}$ have always been used. It is



possible to envisage that analyses of soil erosion could employ instead, for instance, the actual duration of intensity bursts of high intensity, highly erosive rainfall. Analyses might use the length of bursts having intensities of > 80 mm h$^{-1}$ or > 100 mm h$^{-1}$, for instance. These burst characteristics are potentially more closely-linked to processes occurring at the soil surface than the amount of rain in an arbitrary period such as 30 minutes. Thus, if an intensity burst at 80 mm h$^{-1}$ and lasting 15 minutes

occurred within an enclosing rainfall event that was mostly of 4 mm h$^{-1}$ intensity, the I$_{30}$ value, diluted by the inclusion of 15 minutes of low-intensity rain, would be about 42 mm h$^{-1}$. Considering the hour enclosing the intensity burst, the apparent intensity would be just 23 mm h$^{-1}$. This is < 29% of the true intensity of the potentially highly erosive intensity burst. These observations again underscore the potentially serious loss of information that may occur when rainfall data are aggregated to hourly, or even to 30 minute, totals.

4.1 What evidence points to the significance of short-lived intensity bursts?

Multiple studies have shown that short-lived bursts are responsible for much of the runoff and erosion that takes place

on agricultural, dryland, and other soils. Early studies of the importance of intensity peaks embedded within longer rainfalls include the work of Seginer et al. (1962) in Israel. In a study of runoff and soil loss using rainfall simulation, they adopted a 4.5 h design storm that included an intense central 30 min period preceded and followed by lower intensity rain. This central intense period caused most of the observed runoff and erosion. In a study of turbidity and sediment transport in a small Luxembourg catchment, Imeson (1977) logged stream turbidity at 1 min intervals and noted rapid fluctuations on this

timescale. He related this to the sediment contributed by splash of soils from areas close to the channel, reporting that in a 2.2 hour storm with an average intensity of 4-4.5 mm h$^{-1}$, the rainfall arrived as a series of very intense bursts separated by periods of lower intensity. Subsequent work has further explored effects of intensity measured over timescales of a few minutes; only a few examples from a large literature are cited here. Mizugaki et al. (2010) showed that splash dislodgment of soil on Japanese forested hillslopes was strongly related to maximum rainfall intensity assessed at $10 - 30$ minute periods (I$_{10}$ – I$_{30}$). Rainfall

bursts extracted from data tallied over longer periods, such as hourly (I$_{60}$), proved to have less explanatory power. Likewise, Fraser et al. (2011) demonstrated the importance of I$_{15}$ in erosion within the rangelands of northern Australia, and Wagenbrenner & Robichaud (2014) highlighted the importance of I$_{10}$ in post-fire sediment movement in the western USA. At plot scale, Xia et al. (2013) showed that, for cropland in China, I$_{10}$, I$_{20}$, I$_{30}$ or I$_{60}$ were significantly correlated with event losses of N and P, and offered some hypotheses about why one or another index of intensity should become significant at a particular

field site. Nunes et al. (2011) also stressed the explanatory power of I$_{10}$ and I$_{60}$ for erosion under various cover types in Portugal, as did Kiassari et al. (2012) for erosion plots in semiarid Iran. At plot scale, the timing of intensity peaks within a rainfall event has been shown to greatly affect runoff ratios and depths, as well as peak runoff rate (Dunkerley 2012). This effect is thought to arise partly as a result of the intensity peak occurring early, on dry soil, or later in the event when soils have been wetted-up and infiltrability has declined as the wetting front progressed more deeply into the soil.





Even at larger hillslope scale within post-fire landscapes, Wagenbrenner et al. (2014) showed strong correlations between the sediment yield delivered in runoff events and the $I_{10}$ intensity of rainfall. However, as they emphasised, the landscape-scale effects may depend on the relative sizes of the burned area and the area covered by the intense rainfall burst.

Small areas affected by intensity bursts may allow the runoff and sediment to be absorbed or attenuated when the runoff passes over unburned areas or areas not located within the area of intense rainfall. Hubbert et al. (2012) reported $I_{10}$ $I_{30}$ and $I_{60}$ for field sites in southern California in the aftermath of chaparral fire, and again stressed the importance of intensity bursts, especially as expressed by $I_{10}$. Similarly, Taguas et al. (2010) analysed rainfall, runoff, and sediment yield data from a 6.1 ha Spanish catchment under olive cropping. They found strong correlations between $I_{10}$ and runoff volume, runoff coefficient,

the sediment concentration in runoff, and total sediment load. Correlations with $I_{30}$ were smaller, a result attributed to the rapid hydrologic response of the catchment. For urban flash-flooding arising from catchments draining the hinterland of Genoa (Liguria, Italy), Faccini et al. (2018) stressed the importance of rainfall intensities over periods of $1\,h - 3\,h$, while Papagiannaki et al. (2015) in Greece showed that urban flash flooding was probable for $I_{10} > 22.8$ mm h$^{-1}$ (or $I_{60} > 9.6$ mm h$^{-1}$). For Italy, Esposito et al. (2018) reported flash flood thresholds of $I_{10} > 54$ mm h$^{-1}$ or $I_{60} > 30$ mm h$^{-1}$. In catchments SW Germany, Ruiz-

Villanueva et al. (2012) showed that $I_{30}$ offered less explanatory power than $I_{60}$ for catchments $> 10$ km$^2$, owing to their longer response time (greater smoothing of the hydrologic response). For the tropical Lutzito rainforest catchment (3.3. ha) in Panama, Zimmermann et al. (2014) investigated the overland flow connectivity between hillslopes and the main channel. They stressed that short-term intensity measures such as $I_5 - I_{60}$ were important and also showed that the frequency with which the very short-lived intensity characterised by $I_5$ exceeds the soil saturated hydraulic conductivity ($K_{sat}$) has explanatory power, in the

overland-flow-dominated runoff environment. It is possible that on hillslopes, overland flow paths become established during intensity bursts, and the establishment of connected runoff pathways may then facilitate runoff at lower intensities. This may be partly an effect of the clearing of mobile organic litter, which otherwise reduces flow speed and erodibility (Miyata et al. 2009, Ghahramani et al. 2011, Zhou et al. 2018), and perhaps some scour along flow pathways during the intensity bursts.

## 4.2 By what mechanisms do intensity bursts influence soil surface processes?

Here, a brief account is offered of some of the ways in which intensity bursts drive processes of soil erosion.

In terms of soil detachment by splash, an important mechanism linking intensity bursts to splash dislodgment of particles relates to the depth of ephemeral water ponding on the soil surface. Short bursts of intense rain, lasting perhaps minutes, are capable of exceeding the infiltrability of the soil surface, especially if they occur late in the rainfall event when the wetting front depth is greater and infiltrability is consequently lower (e.g. following the Horton exponential model – see Dunkerley 2018). As a result, during an intensity burst, ephemeral ponding may appear in low-lying parts of the soil surface





not previously ponded, and areas of existing shallow ponding may become deeper and cover larger areas. This is important because rates of soil detachment have been shown to increase rapidly as the water ponding depth increases, related to lateral jetting away from drop impact locations. This effect causes a rapid increase in splash detachment as ponding depths increase up to about 3 drop diameters, equivalent to perhaps 10 mm or so for typical raindrop diameters (Gao et al. 2003). As was

shown by Timmons et al. (1971), drop impacts on ponding throw out splash droplets that are primarily made up of water from the ponding layer; surface tension limits the amount of water from an incident drop that becomes splash. Thus there are two consequences of the ponding effect on splash detachment: during an intensity burst, the area of ponding on the soil surface may increase, so enlarging the area over which active soil particle detachment can occur during the intensity burst. Second, existing areas of shallow ponding may be deepened and so enter the depth range where splash is more intense. Simultaneously,

some ponded areas may begin to show connected flow downslope, thus facilitating the movement of organic litter and soil particles. This suite of processes illustrates something of the level of process understanding required to resolve soil erosion processes at the level of detachment mechanisms. However, the experimental data cited earlier show that typically, the flux of splashed soil shows large increases during intensity bursts.

In addition to ponding, it is probable that other mechanisms at work at the soil surface are strongly activated during intensity bursts. The breakdown of soil aggregates prepares smaller, more readily transported particles, and this becomes more active with the higher kinetic energy expenditure at the soil surface associated with intensity bursts (Ewane & Lee 2016). Additionally, air entrapment arises when drop bombardment of the soil surface forces air into the soil pore spaces. Thus reduces available infiltration pathways and by restricting infiltration rates, further encourages ephemeral surface ponding (Wang et al.

20  2014).

4.3 Short-term intensity bursts and climate change

Finally it is appropriate to consider the possible changes in the occurrence of short-lived intensity bursts in a future, warmer climate. Climate change is widely considered to be likely to result in more frequent extremes of rainfall, related in part to the increased moisture capacity of the atmosphere following the Clausius-Clapeyron relation, which suggests an increased moisture-holding capacity in association with warming of the atmosphere of ∼ 7%/°C (Fujibe 2016). Wasko & Sharma (2015) reported that within sequences of rain hours, the most intense showed positive scaling with temperature, and the less intense, negative scaling. However, intensity bursts are an aspect of the intensity fluctuation that is characteristic of most rainfall events

(Dunkerley 2015) and it is not clear how the behaviour of these bursts – lasting only minutes or tens of minutes - might scale with temperature.

There have been multiple investigations of temporal trends in daily and hourly rainfall, based on time-series of historical observations. Not all have found evidence of increasing rainfall extremes. Muschinki & Katz (2013) were able to





find a significant rising trend for only one of 13 stations across the USA, primarily employing data rom 1940-1999. Using data from 1942-2002, Soro et al. (2017) reported mostly decreasing trends in rainfall extremes for the Cote d'Ivoire. Barbero et al. (2017), using data from a large set of US data, found increasing annual maximum daily precipitation compatible with the expected CC scaling (they found ~ 6.9% °C$^{-1}$) but only a lesser scaling for hourly data. Fu et al. (2016) established a significant

trend of increasing hourly rainfall over South China, using 31 years (1982-2012) of data at a large network of precipitation stations. They showed that an increasing frequency of rainfalls accounted for most of the observed effect, with only about 10% attributable to changes in hourly intensity. Sun et al. (2017) analysed sub-daily rainfall (1 h – 24 h) using 442 Chinese stations spanning 1960-2014, and showed complex regional variations in the trend to increasing or decreasing hourly extreme rainfalls.

10        A common approach to the analysis of extreme rainfall intensities is the use of downscaling from daily maxima to sub-daily values, though this is normally only extended to the prediction of hourly maxima. Sub-hourly rainfall data are generated in some canonical cascade downscaling models (Kianfar et al. 2016, Haddad & Rahman 2014, Paschalis et al. 2014, Pohle et al. 2018) though with only moderate success in generating realistic intensities. As noted by Forestieri et al. (2018) downscaling can be undertaken by using correlations between daily and sub-daily maxima established from historical data, on

the presumption that the correlations will remain invariant under climate change. Considering the probable effects of climate change for Sicily, via downscaling from a regional climate model (RCM), these authors derive estimated 1 h rainfall amounts for 50 year return period of 42.05 mm (1972-2003 historical data), increasing by nearly 60% (to 67.1 mm) for 2005-2050. For the USA, using quality-controlled empirical data, Barbero et al. (2017) found that the trend of rising annual daily maximum precipitation was approximately 7% °C$^{-1}$, but only lower rates of change (~ 4% °C$^{-1}$) for hourly extremes. In contrast, Chan et

al. (2016) employed high-resolution convective-permitting RCM predictions of rainfall extremes down to 10 min timescales, for the UK. They found that changes to modelled extremes for 10 min were very similar to those for 1 h duration. They note the need for observational data with which to better test model predictions. Given the importance of hourly sub-hourly precipitation for urban flooding, they argue for the acquisition of additional data on sub-hourly rainfall. Blenkinsop et al. (2018) have described the INTENSE project which is gathering such high-resolution rainfall observational data globally.

Others (e.g. Prein et al. 2016) have argued that moist and dry environments exhibit different temperature scaling relationships, as a function of whether moisture is abundant or limiting. Evidently, much remains to be learned about sub-hourly rainfall and particularly how it may change globally and regional under the predicted warmer climates of coming decades.

The intermittency of rainfall within rain days at the present field site was noted earlier. This is another relatively under-studied

aspect of rainfall climatology (Dunkerley 2015).  Schleiss (2018) demonstrated that failing to allow for the extent of intermittency of rainfall may lead to biased (under-) estimates of the strength of the CC scaling, and suggested a means to untangle the effects of temperature and intermittency on rainfall extremes. It has therefore to be remembered that extreme daily or hourly rainfall amounts are not the same as extreme daily or hourly intensities, given that intermittency results in the daily or hourly rainfall amount being delivered in only a fraction of a day or hour.



## 5. Conclusions

The brief snapshot of soil erosion mechanisms presented above shows that short-lived intensity bursts in rainfall exert an important influence on soil dislodgment and transport mechanisms. The intensity bursts are reasoned to act through several processes, not all of which are as yet completely understood, including effects on ponding depths at the soil surface, the breakdown of soil aggregates into smaller and more readily mobile particles, the establishment and linking of overland flow

pathways, and probably the effects of air entrapment in reducing infiltration rates.

It would be valuable to have the capacity to predict trends in future soil erosion rates. It must be emphasised that this is a complex problem. Several examples of the role of intensity bursts cited above related to post-fire landscapes. Under future climates, many aspects of the factors influencing post-fire hillslope and channel erosion and their spatio-temporal extent may

change. Not least among these is the foreshadowed increase in fire occurrence under future climates, which, through accelerated carbon loss, may feed back into further climate change. Rainfall intensity maxima thus form just one important component of a multi-faceted challenge to building informed land and environmental management for coming decades.

It is important to consider how intensity bursts might evolve in future climates, for which increased erosion risk has

been foreshadowed. However, even contemporary observational data are too coarsely-aggregated to permit the identification of intensity bursts. Therefore, we need additional data for two reasons. First, it will be important to explore more thoroughly the nature of intensity bursts in rainfall in different environments (arid, subhumid, wet tropical, etc.). This would then facilitate a more complete field-based and experimental analysis of how soil erosion rates relate to intensity bursts. Second, a knowledge of intensity bursts can feed into and support attempts to develop improved downscaling methods from global and regional

climate models, by drawing attention to aspects of sub-daily rainfall that are important in fields outside climate science. Contemporary observational data on short-term intensity bursts are desirable for parameterising and validating attempts to downscale predicted daily or hourly rainfalls to the important sub-hourly levels of aggregation, for which $I_{30}$ or $I_{10}$ provide suitable indices. Rainfall data with high temporal resolution should ideally become more widely available to support research in soil erosion, flash flooding, and related areas.

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
