# Peer review of "Rainfall intensity bursts and the erosion of soils: an analysis highlighting the need for high temporal resolution rainfall data for research under current and future climates."

_Earth Surface Dynamics, 2018_

## Short Comment (SC1) · 30 Jan 2019

General: The paper deals with rainfall bursts and focusses on a literature review, pointing out the need for further rainfall studies on finer temporal resolutions (<1 hour) and an enhanced understanding of the physical processes causing erosion. Therefore, the author provides data examples from two study sites highlighting the rainfall bursts, which can only be detected on a fine temporal resolution of <1 hour. I read the paper because i) I was interested in the findings and ii) cascade models are mentioned as disaggregation methods in the manuscript. Hence, my comments cover only parts of

the manuscript dealing with these two topics.

P4l8-9 The author applies a threshold intensity of 30 mm/h for the identification of "intensity bursts" and refer to Tokay and Short (1996). However, the threshold suggested by Tokay and Short is 20 mm/h. For me it is not clear, if this is only a spelling error or the wrong reference. If it is none of both, how was the threshold of 30 mm/h determined? Is this threshold as an absolute value representative for different study sites or does it make sense to express the threshold as a quantile to determine different thresholds for different regions? What is the authors opinion?

P 21l12-13 The author mentions cascade models for rainfall disaggregation and applies the term "canonical cascade model". Only a few paper consider this term as a group name for the family of cascade models. It is more common to distinguish between micro-canonical (rainfall amount is preserved exactly in each disaggregation step) and canonical cascade models (rainfall amount is preserved on average in each disaggregation step), see Schertezer and Lovejoy (1987). Indeed, all the cited references (Paschalis et al. (2014), Kianfar et al. (2016), Pohle et al. (2018)) belong to the group of micro-canonical cascade models. To avoid confusion and misinterpretation I suggest i) to change the term to "micro-canonical cascade models" or ii) to extend the sentence to micro-canonical and canonical cascade models and to provide examples as well for the canonical cascade models.

P21l12-13 In the given reference Haddad & Rahman (2014) no cascade model is applied for downscaling. In the context of erosion processes, I suggest to implement the study of Jebari et al. (2012) instead.

P21 section 2 I would extend the disaggregation part by at least mentioning other disaggregation techniques (e.g. method of fragments, Bartlett-Lewis rectangular pulse models) to increase the awareness of the reader for the existence of different available methods.

P21l13 "with only moderate success in generating realistic intensities". For me, the

term "unrealistic intensities" remains unclear. Since the focus of the study is on rainfall extreme values, I suppose the sentence is about the rainfall extremes identified in the disaggregated time series. However, I would avoid such a general and negative statement There are a couple of investigations where the micro-canonical cascade model is applied for disaggregation processes resulting in a good representation of rainfall extreme values (e.g. in Müller and Haberlandt (2018) for 5-min time steps).

References (additionally to the references already listed in the manuscript):

Jebari, S., Berndtsson, R., Olsson, J. und Bahri, A. (2012). Soil erosion estimation based on rainfall disaggregation. Journal of Hydrology 436-437, 102–110.

Müller, H. und Haberlandt, U. (2018). Temporal rainfall disaggregation using a multi-plicative cascade model for spatial application in urban hydrology. Journal of Hydrology 556, 847-864.

Schertzer, D., and S. Lovejoy (1987). Physical modeling and analysis of rain and clouds by anisotropic scaling multiplicative processes. J. Geophys. Res., 92, 9693– 9714.

---

## Referee Comment (RC1) · Anonymous Referee #1 · 6 Feb 2019

General comments: This discussion paper deals with the issue of rainfall intensity, and how values may be misleading when aggregated on daily or even hourly timescales. Rainfall intensity realistically changes at much shorter timescales on the order of seconds, and high intensity bursts embedded within longer periods of less intense rainfall can be very important for soil erosion. The paper first discusses typical measurement and averaging intervals for rainfall data, and the potential disconnect between these averaged values and features of water erosion of soils and other earth surface processes. The author then introduces an example based on tipping bucket field data from 2 sites,

one arid and the other very wet. The datasets span several years of 1-second reso-
lution data, and the focus here is on inter-tip times (ITT) which provide a reference of
rainfall intensity at the shortest possible time scale. This example detects some very
high intensity bursts at both sites, with several features that vary between sites. Given
an hourly or daily resolution, this data would show very different rainfall intensities from
those observed on short timescales. The paper then finishes with a discussion of how
rainfall influences soil erosion through ponding and other mechanisms, the potential
influence of climate change, and other ways in which rainfall intensity is important to
understand at fine timescales.

Overall, I found this paper to be interesting and important, as it provides an illustrative
example of rainfall intensity at different timescales and promotes a better understanding
of this feature of rainfall. I have several comments listed below on how this paper
could be improved in terms of readability and the presentation of the rainfall analysis.
In the discussion parts of the narrative, the author references many previous studies
on rainfall intensity to explain various aspects of averaging and its importance. This
causes the narrative to often jump from topic to topic, each with a lot of details, and I feel
there could be smoother connections in many discussion paragraphs at the beginning
and end of the paper. I would recommend going through these paragraphs and making
sure there is a main point, that each study cited then promotes. Otherwise, the paper
is nicely written. I also have several minor comments on the presentation of the data
analysis, mainly related to the representation of the results in the table, which are
detailed below.

Specific comments: There are a lot of tables presented, and I think the results would
be more interpretable in Figures instead. For example, the results of Table 1 could
instead be integrated into Figure 1. In Figure 1, the panels should have the same y-
axis ranges to better show differences in intensity, and similarly the x-axis ranges could
be made more clear by using hours instead of Julian day (or in addition to Julian day).
I realize the analysis was done using JD, but it would be much easier to read the plot if

a more typical timescale was used for plotting. The duration of each event, the depths of the events, the mean intensity, and the peak intensity then become more apparent in Figure 1 – with a dotted horizontal line for the mean, a cumulative line for the total depth, etc. Finally, Figure 1 is introduced before the sites, which makes the results a bit confusing – it would be good to briefly mention the different sites in the text just before introducing the figure, and then going into details later.

The paragraph starting on Line 6 of page 6 is very technical, and I think some of this information could be supplementary or unnecessary, as it breaks the flow into the data analysis at the field sites. The MIT definition is important, and the paper could use an additional sentence explaining this at the end.

Page 6 line 20: does N refer to total number of events during the study period? Page 7 line 17: I would add "due to differences in the tipping bucket sizes," at the beginning of the sentence here, as to why the values are different between FG and MM Page 7 line 20: the term "nominated value" is strange – maybe say "a chosen threshold value"? Table 2 and Table 3: a separation (e.g. bold line) would be useful between the third and fourth column. Alternately, as in Figure 1, I think this could be better shown in a figure. For example, each exceedance duration is embedded into the next longer duration, if I am interpreting it correctly, so a sort of cumulative plot could show all the same information more succinctly. The first 3 columns here seem redundant, and could be given with an equation related the ITT, the min intensity, and the tips at that intensity instead of the table.

During the middle part of the paper which focuses on the results, the reader could use some reminders of the acronyms at intervals (e.g. on page 9, line 15, remind us of the site names)

Figure 2: This result could be made more powerful by combining the lines onto a single plot, so the magnitudes could be compared. A logarithmic scale could be used on the y-axis to distinguish the magnitudes, and show that the 1 hour aggregation loses

an order of magnitude in intensity compared to the unaggregated values. Table 4-5: similar comment for this in Figure form instead (a pie chart, bar chart, etc since it is segmenting the ITTs into their time ranges), and a side-by-side comparison between the two sites would be more useful to talk about this feature.

Figure 3: squares and diamonds are mentioned in the caption – but I see circles and triangles in the Figure (a legend in the Figure would be useful) For this Figure and Figure 4, I think Equations 1-5 could be removed and placed inside the appropriate subfigure panels.

Table 7 is referenced in the text but I don't see it in the manuscript?

Page 17 line 14: the phrase "duration have durations of hours" is very confusing, I had to read it a few times before it was clear. Page 17 line 25: "relationship" Page 17 line 22: "hourly"? Page 17 line 29: The "I_30" index is not well-defined here, and this type of index is referred to many times in the discussion part of the paper – would benefit from a definition here.

Page 19: This paragraph is an example of one with a lot of information contained in it. It could use improvement of language to synthesize and highlight the main issue of erosion on post-fire landscapes.

Page 20, line 21: incomplete sentence Page 21 line 29: "present field site" – be more specific here and indicate specific sites In the conclusion section, when you say "the brief snapshot of soil erosion mechanisms" – I feel that the previous sections are actually more broad in terms of not only soil erosion, but climate change, urban flooding, and other aspects of rainfall and climate. I think this indicates that the previous sections could be better tied in with the main topic of rainfall intensity and soil erosion, e.g. in the section on climate change and the CC relationship and how rainfall intensity may be influenced, bring the discussion back to soil erosion. Otherwise, this could be done in the conclusion with a few extra sentences. Additionally in the conclusions section, it would be useful to refer back to the field data analysis in the context of this broader

discussion of how finer resolution rainfall data is needed to link between rainfall and soil erosion.

---

## Referee Comment (RC2) · Anonymous Referee #2 · 25 Feb 2019

**Summary**

This paper presents an argument for measuring rainfall at higher temporal resolution (preferably sub-hourly) than is typically available in contemporary precipitation datasets (where daily or at best hourly rainfall records available) in order to capture "intensity bursts," or short periods of potentially extremely intense rainfall, that may exert a much stronger control on soil removal/transport events than mean rainfall intensity derived from longer averaging periods. A broad overview of erosion and transport mechanisms directly affected by precipitation is given in the introduction and discussion sections,

while the main results of the paper pertain to describing intermittency and the increasing relationship between rainfall intensity and temporal resolution using time series of precipitation measurements from two field sites in Australia. I find the description of intermittency using the two rainfall datasets to be rather qualitative in nature. The author cites several studies (e.g. Paschalis et al., 2014; Monjo, 2016; Beranova et al., 2018; Pohle et al., 2018; Schleiss, 2018) that provide a more quantitative view of intermittency (using, e.g., multifractal analysis), but chooses not to apply the tools used in those studies in this manuscript. These analytical techniques are sufficiently complex that I understand the author may not have wished to complicate the study with a lengthy discussion of the relevant mathematical framework, although I do think the author's argument would be more convincing with a more rigorous quantification of intermittency or the impact of intermittency on predicted erosion and transport rates.

While I agree with the author's primary thesis that using precipitation measurements at hourly or daily aggregation leads to significant underestimation of process rates relevant to erosion or sediment transport, I don't see a specific suggestion for how precipitation should be measured to more accurately represent the physics at play. My main concern is that temporal intermittency is addressed without paying comparable attention to spatial intermittency. In essence, I don't see why having greater temporal resolution from point measurements (be it from rain gauge, disdrometer, etc.) necessarily leads to a better understanding of plot or catchment scale erosion/sediment transport. Even over an extremely homogeneous land surface, intermittency is generated in clouds aloft due to their close coupling with atmospheric turbulence, and surface rainfall likewise exhibits significant intermittency. Accounting for this spatial intermittency seems just as important the temporal aspect; indeed, this very issue has been explored extensively by the hydrology community (e.g. Berne et al., 2004). I recognize that such a broad study was not the point of the present article; I bring all of this up primarily so the author is aware that there exists a large body of work relating to these questions.

**ESurfD**
Overall, the paper attains its stated purpose: to illustrate the intermittency of precipitation (i.e. the magnitude of rainfall intensity maxima increase as measurements are disaggregated) and highlight the potential impacts on earth surface processes. I have some concerns about the presentation of results, but I can without hesitation recommend this article for publication with minor revisions.

Please find below other general comments, followed by specific comments. Specific comments refer to "Pxx, Lyy" where P is followed by a page number and L by line number.

General comments:

\* A major question for me regarding the measurements presented is whether buckets with different tipping volumes can be directly compared using "inter-tip time." The manuscript indicates that "raw" ITTs are analyzed – is any normalization performed to account for different bucket volume? It's not quite comparing apples and oranges (perhaps more akin to apples and pears), but I think it is important to quantify what the impact on ITT run statistics would be if, say, MM had a 0.5 mm capacity instead of 0.2 mm to understand whether bucket volume affects diagnosed intermittency.

\* Modified Julian dates are convenient for running the analysis, but not for conveying the results. Consider plotting a more intuitive time format (i.e., HH:MM:SS with yyyy/mm/dd date below) so the reader can easily see that sub-hourly variability is important. This is not to say that your use of MJD is misguided, simply that the points you are trying to convey will be "punchier" if readers don't have to spend mental effort decoding what a fraction of an MJD means.

\* Regarding the use of tables in Section 2.2: Tables 2 and 3 are very difficult to understand in their present form. There are 7 columns of data and comparing the two very large tables to understand differences between the MM and FG sites is nontrivial. I would prefer to see these data presented graphically, perhaps even in one unified loglog axis figure. This would also enable you to reduce the amount of data presented, **ESurfD**
e.g. "number of tips in longest run" and "rain depth of longest run" are related by a simple multiplicative constant, and number of tips is subject to the bucket volume issue discussed above such that I don't think the two sites can be easily and directly compared using that metric. Conversely, the statistics given in the first two paragraphs of this section (average rain day accumulation and intensity, hourly intensity, various percentile values of intensity, etc.) would be well suited to tabular presentation, especially considering that one of the author's main points is to contrast conditions at the two sites considered. Regarding the percentile values, the cumulative distribution functions of rainfall intensity at hourly/daily aggregation could also be presented in graphical format for ease of comparison.

\* The threshold given for "extreme" precipitation in Tokay and Short (1996) is 20 mm h-1 as opposed to the 30 mm h-1 value used in the manuscript. While I suspect the exact choice of what constitutes "extreme" intensity is likely somewhat unimportant, what fraction of MM and FG rain at 1 s resolution falls in this category using the Tokay and Short definition of R>20 mm h-1? Is there a reason you used a modified criterion?

\* What is the end goal of increasing temporal resolution? Is it to resolve individual raindrop collisions with the ground? Or to have some "ideal" sampling of the rain rate distribution (the definition of "ideal" being a subjective determination)? Or something else entirely? As stated above, I agree with the author that hourly/daily aggregation is of limited relevance for characterizing erosion, or really any other process that is highly nonlinear and therefore driven by extreme values, but there are other considerations that inform how temporal resolution is chosen for precipitation rate measurements than aggregation for the convenience of reducing data amount. In this context, it may be seen as a tradeoff: having increased knowledge of the "instantaneous" rain rate at a single "point" or rain gauge may be offset by less confidence that the instantaneous measurement at a given gauge is representative of the rain rate over some broader spatial area of interest (e.g. plot scale) precisely because of the intermittency aspect of rain.

**ESurfD**
\* Section 4.1 is a great argument for having higher resolution rain rate measurements, but there's still quite a disconnect between the 5-10 minute fixed clock intensities cited here versus the 1 s measurements presented. Is this intentional?

Specific comments:

P3, L22: Peters and Christensen (2002) citation not in bibliography

P5, L19/23: Are the "mean annual rainfall" figures given simply the average over the observational records analyzed in the manuscript? If so, please explicitly state that the averages are over the observational period and not derived from some longer term record. It would be inaccurate to frame these as climatological means, especially given the author's comments regarding the large variability of annual rainfall at the FG site.

P6, L16: What is the sensitivity of the results to the choice of MIT? Is a single value of MIT valid for two different sites? What I'm getting at is that you've set this number as if it were a universal constant, while different storm systems may have different characteristic inter-event times. In any case, this is more something that I'm curious about than a comment that calls for rigorous sensitivity testing.

P9, L15-19: The data presented here would be more digestible if also given in a table.

P10, Figure 2: Can these three curves be presented with differing line style or color in a single figure? It would starkly highlight the loss of magnitude due to coarse-graining the measurements thereby giving greater weight to your argument. If this is undesirable, I am interested to hear your argument as to why.

P11-12, Tables 4-5: As with Tables 2-3, this data could be much more concisely expressed in graphical form.

P17, L25: Typo, "relatiobship" approximately halfway through line.

P19, L6: Insert commas in "Hubbert...reported I10 I30 and I60 for..."

P19, L14: "In catchments in SW Germany" (add "in" after "catchments")
**ESurfD**

---

## Author Comment (AC1) · 11 Mar 2019

**David Dunkerley**

david.dunkerley@monash.edu

Received and published: 11 March 2019

I much appreciate the comments provided on the manuscript by Dr Müller-Thomy. I have adopted the suggestion to refer to the papers cited as exemplifying microcanonical cascade models, rather than simply as 'canonical cascade models'. I have also included the paper by Jebari et al. (2012) in place of Haddad and Rahman (2014). I have added the reference to Müller and Haberlandt (2018) and have referred to their findings about the errors in over-estimation of short-term intensities. I hope that these revisions will make the paper clearer and more informative.

**ESurfD**

---

## Author Comment (AC2) · 11 Mar 2019

I sincerely thank both anonymous referees for undertaking reviews of my manuscript and for their insightful comments. I hope that I have been able to improve the paper in light of these comments. In the following, I have attempted to identify the principal comments in order, and provide a response to each.

Referee 1

C: Discussion could be more smoothly connected to the topic of the paper. R: I have

been through these sections, and added some linking text that I hope will maintain the line of argument more clearly. A few new references have been added in the course of these revisions.

C: There are a lot of tables presented and I think the results would be more interpretable in Figures instead. R: To reduce the amount of data presented in Table form, I have eliminated Tables 2, 3, 4, and 5, and replaced these with Figures containing the relevant data. The new Figures are Fig. 2 and Fig. 5 in the revised numbering.

C: Paragraph starting line 6 of page 6 is very technical. MIT needs a definition. R: Some of the technical detail of data processing, including the SOFA subroutines used, has been deleted in order to simplify this presentation. I have added a definition of minimum inter-event time (MIT) and added an earlier paper (Dunkerley 2008) on this subject.

C: Does N refer to the total number of events during the study period? (and related comments). R: Yes. I have added an additional clarification of this point, and likewise in reference to the differing bucket size in the rain gauges at the two field sites.

C: The reader could use reminders of the acronyms used. R: I have added these as suggested.

C: regarding elimination of Tables 4 and 5. R: these Tables have been removed, and replaced by a new Figure.

C: error in caption referring to squares and diamonds. R: this caption has been corrected.

C: reference to missing Table 7. R: this reference has been replaced with the correct reference to what is now Table 2 in the revised numbering.

C: typographical and other errors. R: all minor errors have been corrected.

C: need for some additional focus on soil erosion in Discussion or Conclusions. R:

Some additional discussion on this point has been incorporated into revised Conclusions, highlighting the focus on soil erosion.

---

## Author Comment (AC3) · 11 Mar 2019

**David Dunkerley**

david.dunkerley@monash.edu

Received and published: 11 March 2019

I sincerely thank both anonymous referees for undertaking reviews of my manuscript and for their insightful comments. I hope that I have been able to improve the paper in light of these comments. In the following, I have attempted to identify the principal comments in order, and provide a response to each.

Comments from Referee 2

C: spatial intermittency should be considered as well as temporal intermittency R: I

thank the referee for this comment. I had not neglected spatial issues, but eliminated any discussion of this (apart from the section discussing post-fire erosion, where the spatial extent of intense rain is highlighted) as being beyond the scope of the paper. However, I have inserted some additional text to highlight this issue in a new Section 4.4 in the Discussion of the revised paper.

C: Comparing ITTs with buckets of differing capacity. R: It would certainly be preferable had both tipping buckets had the same capacity. However, the FG site was set up more than 17 years ago, and the rain gauge installed at that site, which has been retained since its installation, had a 0.5 mm bucket capacity. I have tried to make the difference in data resolution clear in the discussion of ITTs, in which shorter ITTs can sensibly be analysed for the MM site (0.2 mm bucket capacity) than at the FG site. The different sensitivities at the two field sites is certainly not ideal.

C: Modified Julian dates (MJDs) are unhelpful to read in Figures etc. R: It is very difficult to represent intensity variations using Gregorian calendar dates with the resolution that can easily be achieved using MJDs. Gregorian dates would need to specify separately year, month, day, hour, minute, and second at tick marks along the time axis of Figures, which is difficult to achieve. Appreciating that MJDs are difficult to interpret, I have modified all of the Figures to show the start and end dates of each rainfall event in the Gregorian system. There are online conversion tools that can also assist here, such as the MDJ to Gregorian converter at http://www.csgnetwork.com/julianmodifdateconv.html.

C: Tables 2 and 3 are difficult to understand. R: As noted above in the response to Referee #1, these Tables have been removed and replaced by Figures.

C: Regarding the threshold of 'extreme' rainfall R: This has been corrected to show the 20 mm h-1 criterion of Tokay & Short (1996).

C: what is the goal of increasing the temporal resolution? R: The argument presented in the paper is that hourly data, for instance, sacrifice too much intensity resolution for studying the influence of intensity on many landsurface processes. My view is that
since event data loggers can readily store unaggregated tip event data, that this should be more widely collected and retained, rather than data that have been temporally aggregated. The sensitivity of tipping-bucket gauges is of course itself limited owing to the time required to fill the buckets, but this provides a good reason for wishing to avoid an additional loss of temporal resolution. The point about the need to also consider spatial resolution is very pertinent, though collecting the required data will be difficult. For specialised field research sites, at least, such as those involving soil runoff and erosion plots, or small fields, a 'point' where the character of the rainfall is known from high-resolution rainfall data, in sufficient detail to resolve intensity bursts, may often be sufficient. For whole catchment studies, the challenge of acquiring adequate spatiotemporal data resolution is a daunting one.

C: minor errors: Missing Peters & Chistensen (2002) reference etc. R: all of the minor errors listed have been corrected; the Peters and Christensen reference has been added.

C: Can the three graphs of original Fig. 2 be combined into a single Figure? R: This seems an excellent suggestion, but having attempted to do this, I found that it was not really achievable. The three intensity peaks are coincident in time, and this causes the curves to lie closely on top of each other. Additionally, the intensity of the data aggregated to 1 h are very small (commonly less than 2 mm h-1) and on the vertical scale required to show the unaggregated data (which extends to > 200 mm h-1) the 1 h data become unreadable. Using a log scale for the intensity axis radically alters the apparent character of the intensity brusts. I have therefore retained the layout of the original Figure, but with the Gregorian dates added to the time access (in addition to the Modified Julian Day numbers) as an aid to readability.

**ESurfD**